# Enhanced Production of Biogas Using Biochar–Sulfur Composite in the Methane Fermentation Process

**DOI:** 10.3390/ma15134517

**Published:** 2022-06-27

**Authors:** Ewa Syguła, Michalina Gałęzowska, Andrzej Białowiec

**Affiliations:** Department of Applied Bioeconomy, Wrocław University of Environmental and Life Sciences, 37a Chełmońskiego Str., 51-630 Wrocław, Poland; ewa.sygula@upwr.edu.pl (E.S.); 116119@student.upwr.edu.pl (M.G.)

**Keywords:** sulfur, biogas, biochar, methane fermentation

## Abstract

The methane fermentation of organic waste is one way to minimize organic waste, which accounts for 77% of the global municipal waste stream. The use of biochar as an additive for methane fermentation has been shown to increase the production potential of biogas. Sulfur waste has a potential application to synergistic recycling in a form of composites with other materials including biochar. A composite product in the form of a mixture of biochar and molten sulfur has been proposed. In this experiment, additions of the sulfur–biochar composite (SBC) were tested to improve the fermentation process. The biochar was produced from apple chips under the temperature of 500 °C. The ground biochar and sulfur (<1 mm particle size) were mixed in the proportion of 40% biochar and 60% sulfur and heated to 140 °C for sulfur melting. After cooling, the solidified composite was ground. The SBC was added in the dose rate of 10% by dry mass of prepared artificial kitchen waste. Wet anaerobic digestion was carried out in the batch reactors under a temperature of 37 °C for 21 days. As an inoculum, the digestate from Bio-Wat Sp. z. o. o., Świdnica, Poland, was used. The results showed that released biogas reached 672 mL × g_vs_^−1^, and the yield was 4% higher than in the variant without the SBC. Kinetics study indicated that the biogas production constant rate reached 0.214 d^−1^ and was 4.4% higher than in the variant without the SBC.

## 1. Introduction

### 1.1. Methane Fermentation with Biochar

In the methane fermentation process, organic waste and biomass (lignocellulosic) can be used [1]. At the moment, organic waste makes up 77% of the municipal waste stream in the world [2]. The European Union has introduced a document regulating the selective collection of biodegradable waste, which must be collected selectively from 2023 [3]. The production of biochar from organic waste with a low lignin content, which guarantees a high specific surface area, is another branch of using waste for purposes other than landfilling [4]. At this time, a significant proportion of organic waste is landfilled. EU legislation requires 10% of organic waste to be landfilled by 2035 [5]. In turn, sulfur waste, which is now utilized, can also be upcycled. As a result, composite SBC (S—sulfur, B—biochar, C—composite) from waste may increase the efficiency of the methane fermentation process. Such actions are in line with the circular economy—recycling the waste stream for reuse and giving it good properties.

The application of methane fermentation technologies for energy production can be an excellent alternative to the uncertainties of regional energy policies. Using biological processes to treat organic materials, which account for 77% of the global municipal waste stream, is an opportunity to exploit the energy potential. During methane fermentation, organic material is decomposed under anaerobic conditions by anaerobic microorganisms at 37 °C or 55 °C [6].

Anaerobic digestion (AD) technology is a widely used method of waste treatment [7]. Current research focuses on increasing the efficiency of biogas production while respecting the principles of the circular economy. One example is the use of biochar from organic waste in thermal torrefaction and pyrolysis under anaerobic conditions [2,8]. Świechowski and collaborators have shown that the addition of biochar can improve methane yield in the AD of organic waste by 3.6% [3]. In contrast, Lou showed an improvement in methane production compared to a sample without the addition of biochar. In these tests, glucose was used as a substrate; therefore, the efficiency of the process was much higher and even amounted to 86.6% compared to the control sample—without biochar. [9]. Biochar has been found to help improve methane production, but it is not clear what direct effects this has on microorganisms. It can be assumed that the high porosity of biochar has a positive effect on adhesion surfaces and the increased populations of methanogenic organisms. In the fermentation process, biochar has many functions, including maintaining a stable pH, facilitating the direct interspecies transfer of electrons, and promoting the growth of microorganisms. In addition, the sorption properties of biochar contribute to the reduction in CO_2_ and H_2_S and other potentially toxic byproducts. A significant specific surface area is favorable for the development of colonies of microorganisms [10]. Compounds such as non-organic nitrogen, long-chain fatty acids, and sulfides harm the AD process. However, new types of biochar and biochar-based composite materials used for the enhancement of AD are being developed. The heterogenous structure and properties of composite materials may create a multifunctional niche for the simultaneous development of different groups of microorganisms. Therefore, for the first-time, we propose combining biochar with elemental sulfur to achieve the synergistic effect of biochar and sulfur properties. The addition of sulfur to the fermentation process seems controversial; however, preparing the material in advance and keeping it in the right conditions may bring benefits. Methane fermentation is a favorable process to produce hydrogen—the fuel of the future. Accordingly, all forms of process efficiency improvement are part of building knowledge about new aspiring fuels [11].

### 1.2. Production of Sulfur

Emission standards have been introduced because of an increasingly conscious approach to the environment. These standards vary according to the region and the level of development of the country [12]. Fossil fuel combustion continues to account for the largest share of global energy production. The combustion of fossil fuels generates significant emissions of gases such as SO_x_. The exhaust gases from the combustion of fossil fuels are desulfurized, resulting in waste sulfur. Sulfur can also be removed before combustion. This method is commonly used for liquid and gaseous fuels. For example, solid fuels such as coal may have a sulfur content of up to 14% in liquid fuels, while gases may have a sulfur content of up to 5.5% [13]. Sulfur can be produced in two main processes: the extraction of sulfur from deposits and the desulfurization of fuels such as petroleum and natural gas. However, emission standards have made the extraction of sulfur uneconomic, as waste sulfur is available in large quantities and at low prices.

### 1.3. Composite Materials

Sulfur in construction is often used as a component to improve the strength properties of products [14]. However, the use of sulfur in commercial construction is not possible due to the low melting point of sulfur. The use of sulfur concrete is possible in elements of road and sewer infrastructure [15]. It should be noted that a sulfur coating in contact with a liquid does not alter the pH of the liquid at a temperature below 135 °C [16]. In addition, sulfur in composite materials has antiseptic and hydrophobic properties, which makes sense when used in sewerage infrastructure. Common concrete, consisting of water additives and cement, leaves a significant carbon footprint. The production of 1 ton of cement emits up to 650 kg of CO_2_. However, when transport, energy, and heat are taken into account for the production and use of cement, emissions can even be as high as 1:1 [17]. Sulfur concrete consists of molten sulfur and aggregate. Compared to concrete, no cement is used, and therefore, no water is used. The use of waste sulfur from the capture of emissions would be an excellent alternative to a closed loop. In addition, crushed sulfur concrete can be reused to produce this product. After single crushing, sulfur concrete showed better mechanical properties than the original product [18].

### 1.4. Properties of Sulfur

The basic nutrients are carbon, hydrogen, oxygen, nitrogen, phosphorus, and sulfur. Without these elements, there is no life or development of organisms. Sulfur is an element that plays an important role in the photosynthesis process, being the basic nutrient of plants and directly influencing the yield of crops. Sulfur has antiseptic properties, supporting resistance to pathogens [19]. The physical properties of sulfur depend on the allotropic form. The two main allotropes of sulfur are orthorhombic sulfur and monoclinic sulfur. Orthorhombic sulfur at the temperature of 95.5 °C goes into the form of monoclinic sulfur, and then at the temperature of 119.3 °C, it changes its state from solid to liquid. The viscosity of liquid sulfur increases with temperature, up to 444.6 °C, at which the sulfur boils [9,20]. Temperature also affects the sulfur density as well as the thermal conductivity, where the solid state of the aggregate shows better insulating properties. The combination of sulfur and hydrogen forms hydrogen sulfide, while in the combustion process, sulfur dioxide formed in combination with water forms sulfuric acid [21].

### 1.5. Aim of the Study

Due to the growing interest in the application of biochar to the methane fermentation process, as an alternative upgraded solution, the application of a sulfur–biochar composite (SBC) was proposed. The addition of sulfur to the methane fermentation process may be controversial, but the fermentation process takes place at a temperature lower than the sulfur melting point [21,22]. As a result of the above, there is no risk of changing the pH of the environment. On the other hand, it is hypothesized that the new structure of the SBC may influence the production of biogas. Therefore, the aim of this preliminary experiment was to investigate if the new type of composite (SBC) may improve biogas yield.

## 2. Materials and Methods

### 2.1. Materials

#### 2.1.1. Inoculum

As an inoculum for the experiment, digestate from an agricultural biogas plant was used (Bio-Wat Sp. Z. o. o., Świdnica, Poland). The biogas plant carried out the AD process in wet and mesophilic conditions. The digestate was collected from a post-fermentation chamber and delivered to the laboratory, where it was filtered to remove solid contaminants.

#### 2.1.2. Kitchen Waste

Kitchen waste was prepared according to the recipe of Świechowski and collaborators [23]. The waste consisted of vegetables, 41.6% (lettuce, potatoes, and carrots, each 13.86%), banana peels, 29.7%, basic food, 22.3% (pasta, rice, and bread, each 7.43%), chicken, 0.2%, eggshell, 4%, and walnut shell, 2.2%. Food waste was ground and dried for 24 h at 105 °C. The moisture content of the kitchen waste was 42.5%, and the dry organic matter content was 95.8%.

#### 2.1.3. Biochar

The biochar used to produce SBC was made of applewood chips. The process of the low-temperature pyrolysis of apple chips was carried out in a muffle furnace (SNOL, 8.1/1100, Utena, Lithuania) at a temperature of 500 °C and with a processing time of 1 h. After the process, the biochar was ground to a powder form to obtain the same granulation of the material as the waste sulfur.

#### 2.1.4. Sulfur

For biochar sulfur composite production, the crystal sulfur was provided by ORLEN Poland Group S.A. The preparation of the material for the tests consisted in crushing the crystal into smaller agglomerates. The material was then ground to a powder in a laboratory mortar.

### 2.2. Methods

#### 2.2.1. Production SBC

The ground biochar and sulfur were sieved separately through a 1 mm mesh sieve to homogenize the material. Then, fractions below <1 mm were mixed. Subsequently, the mixture of biochar and sulfur was put into a silicone mold which was heated for 2 h at a temperature of 140 °C. The weight ratio of both substrates was 60% sulfur and 40% biochar. After the process, the SBC was removed from the furnace and ground using a laboratory mortar.

#### 2.2.2. Anaerobic Digestion

The biogas potential test was performed using the OxiTop measurement system (WTW, Weilheim, Germany). The system consisted of glass bottles—fermentation reactors and manometric heads calibrated to measure the pressure difference due to biogas production. The measurement was read continuously with the OxiTop OC 100 controller (WTW, Weilheim, Germany). A side connection was used to drain excess gas from the reactor. The reactors were kept in a climate chamber at 37 °C to ensure mesophilic conditions (Pollab, Wilkowice, Poland).

The process was set up for the digestate only, digested with the addition of kitchen waste (substrate) and digested with substrate and the addition of 1% by the weight of the SBC. The diagram of the procedure is presented in Figure 1.

The water content in the digestate was 94.91%, and in the substrate, it was 42.45%. The dry organic matter content in the digestate was 8.34%, and in the substrate, it was 95.80%. The Table 1 shows the exact amounts of the individual components in the individual reactors. Each mix was set up in triplicate.

The duration of the AD process was 4 weeks. Data showing the change in pressure over time were downloaded to a computer and then analyzed. Based on the collected data, the free capacity of the reactor and the number of biogas moles were determined, and the production potential was converted into volatile solids (vs). Then, in the Statistica 13 software (StatSoft, Inc., TIBCO Software Inc., Palo Alto, CA, USA), first-order reaction models were evaluated, and the biogas production kinetics were calculated. The following formulas were used to calculate the biogas kinetic parameters [3,6]:(1)$$B_{t}=B_{0}\times {\left(1-e\right)}^{\left(-k\cdot t\right)}$$
where:B_t_—biogas volume over time t, mL × g_vs_^−1^;B_0_—maximum production of biogas from the substrate, mL × g_vs_^−1^;K—reaction constant rate, d^−1^;t—process time, days.
(2)$$r=B_{0}\times k$$
where:r—biogas production rate, mL × g_vs_^−1^ × d^−1^.

## 3. Results and Discussion

### 3.1. Production of SBC

As a result of the thermal process of the biochar and sulfur mixtures in a muffle furnace, the SBC material was obtained. As a result of being heated to 140 °C, the sulfur turned into a liquid in the formula, and it was a binder in the composite material. The filler was biochar in the amount of 40% of the weight of the material. The process resulted in a solid product, the sulfur was completely dissolved, and no agglomerations of undissolved sulfur were visible. Figure 2 shows the biochar, sulfur, and SBC (60% sulfur and 40% biochar from applewood chips).

### 3.2. Anaerobic Digestion

The collected data were recalculated and are shown in Figure 3. Each research sample was performed in triplicate. The graph shows the mean results of three different mixes. Reactors with the digestate alone showed a low potential for biogas production, not exceeding 100 mL × g_vs_^−1^. Anaerobic bacteria did not release large amounts of methane due to the lack of organic matter. In the remaining cases, the amount of released biogas reached 657 mL × g_vs_^−1^ for reactor IKW_SBC (I) and 672 mL × g_vs_^−1^ for IKW_SBC (II).

Figure 3 shows that the addition of 1% by the weight of the SBC with a 60% S share increased biogas production compared to the digestate used with kitchen waste. Only during the first measurement was the biogas production higher for reactor IKW(I); with each subsequent entry, the reactors with the addition of the SBC showed higher separated amounts of biogas. On the first day, reactors IKW_SBC (I) and IKW_SBC (II) produced 19% more biogas than reactor IKW(I). On days two to four, it was already 6% more, while on each subsequent day of the process, 2–4% more methane was produced. When counting the arithmetic mean for four weeks, it was found that the addition of 1% of the weight of the sulfur–biocarbon composite to the weight of kitchen waste resulted in an increase in biogas production by 4%. This value seems to be relatively low; however, it is comparable with other results of the application of biochar alone, as manifested by an improved biogas yield of 8% [24], 6.6% [25], or 5% [6]. The improvement in the applicability of the SBC requires further investigation, including a wider temperature range of biochar production, different feedstocks for pyrolysis, different proportions between biochar and sulfur, and new methods of binding biochar with sulfur.

There are many mathematical models used to test the biogas production potential in the AD process. It has not yet been determined which models best reflect the experimental data. The result is an overlapping of many variable factors (type of substrate, chemical reactions taking place, or their concentration). Many authors, however, agree with the choice of the first-order kinetic model that was used for this test [26].

The highest values of each kinetic parameter were obtained for the reactor with the addition of the SBC. Higher values translate into more biogas production at a faster rate. The highest constant of the biogas production rate (k) was in sample IKW_SBC, which was 0.214 d^−1^, while reactors without the SBC had a k value of 0.205 d^−1^ (Table 2). The maximum production of biogas from the substrate (B_0_) for the IKW and IKW_SBC reactors differed by 1 and 2 mL × g_vs_^−1^. The rate of biogas production (r) in reactor IKW_SBC was 6% higher than in reactor IKW. The determination coefficients (R^2^) showed a high level of adjustment of the measurements to the model, at the level of >0.98.

The innovation in this research consists of the use of a potential process inhibitor (S) as an additive improving the efficiency of the process. Thanks to the thermal treatment of sulfur and biochar, the properties of the product under methane fermentation conditions did not disturb the process, which was confirmed by the increase in biogas efficiency. However, no clear reason for the increase in the amount of biogas was determined, and the quality of the resulting biogas was not determined. Research should be continued, as there is a basis for extending it.

## 4. Conclusions

The produced SBC material was tested for the addition of methane fermentation. It was shown that the yield of biogas production increased compared to the control samples. The implementation of SBC in the fermentation reactor increased the production by 4% and the maximum methane yield by 18 mL × g_vs_^−1^. The proposed first-order reaction models corresponded well with the experimental data. The correlation coefficient—R^2^ for all three samples ranged from 0.98 to 0.99. The fermentation process was optimal, as almost the maximum values were obtained. Waste was used for the research, which does not apply to any economic sector. A significant portion of the stream is recycled, and the disposal of transformed sulfur in oceans has even been proposed, with the awareness that the ecosystem would be disturbed by this. Waste sulfur after transformation into composite shows hydrophobic properties, while other authors have shown that the contact of the material with water does not affect the pH. Our demonstration of an increase in biogas production with the SBC material opens up opportunities to improve production efficiency. However, the implementation of such procedures should be preceded by detailed research. Quantitative studies have been carried out; the next step is to define the qualitative aspects of the implementation of the SBC into the reactor.

## Figures and Tables

**Figure 1 materials-15-04517-f001:**
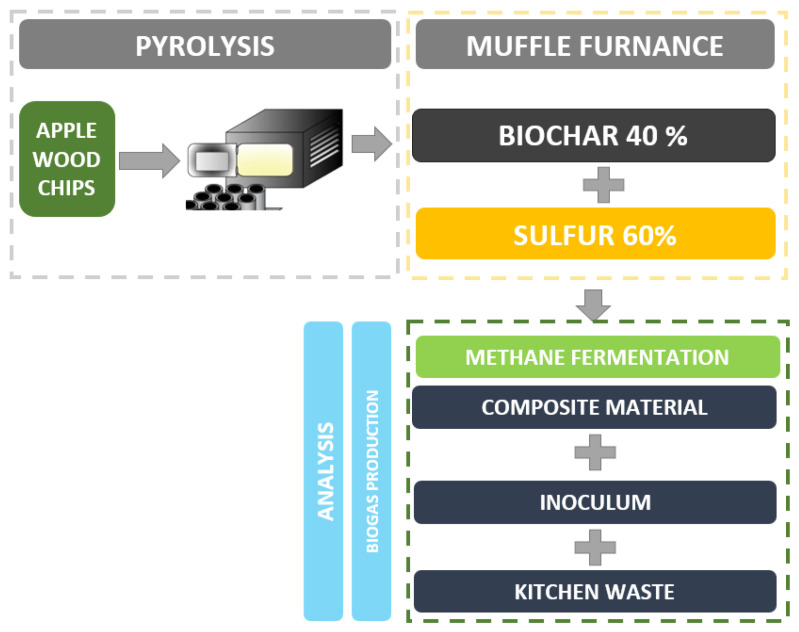
Procedure to produce biogas and SBC mixture to test the potential of biogas production.

**Figure 2 materials-15-04517-f002:**
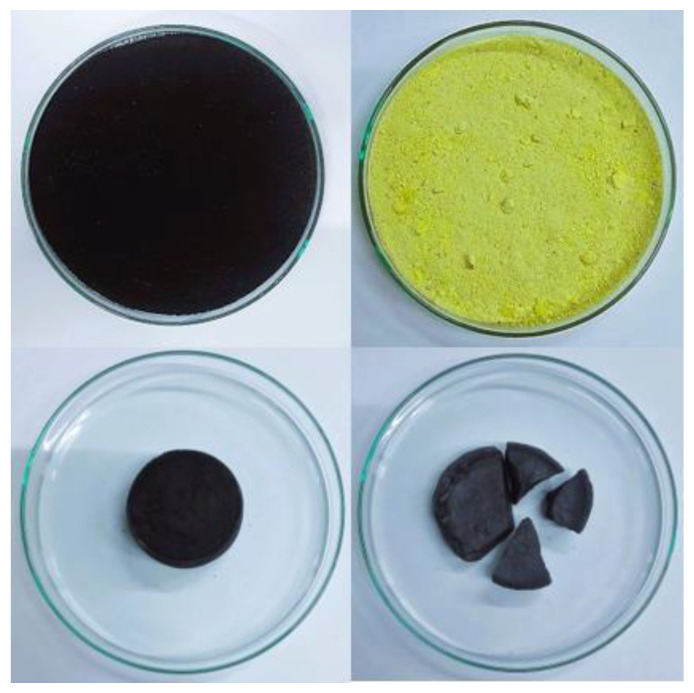
The biochar from applewood chips (**left**, **upper corner**); sulfur (**right**, **upper corner**); SBC (**bottom**)—samples located in the Petri ditches with the diameters of 15 cm.

**Figure 3 materials-15-04517-f003:**
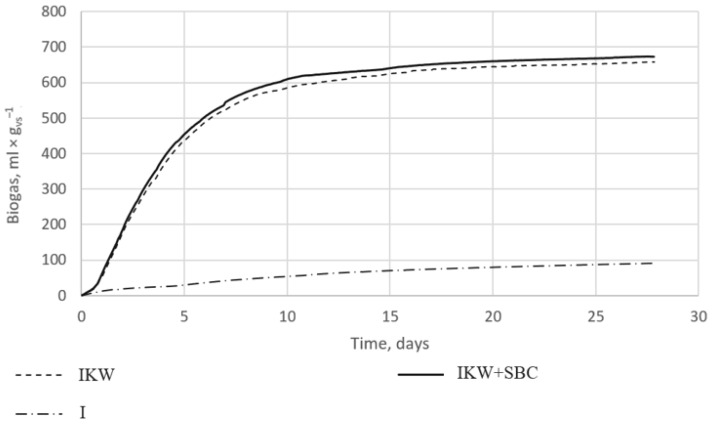
Graph of biogas growth curves over time for the control sample—digestate (I), digestate + substrate (IKW), and for digestate + substrate + SBC (IKW + SBC).

**Table 1 materials-15-04517-t001:** Masses of individual components in methane fermentation reactors (I—inoculum, IKW—inoculum with kitchen waste, SBC—sulfur–biochar composite).

Reactor ID	Inoculum, g	Kitchen Waste, g	SBC, g
I	200.00	-	-
I	200.00	-	-
IKW	200.00	2.500	-
IKW	200.00	2.502	-
IKW + SBC (I)	200.00	2.510	0.2509
IKW+ SBC (II)	200.00	2.509	0.2506

**Table 2 materials-15-04517-t002:** Kinetic parameters of AD.

Reactor Content	Parameter	Values	Unit
**I**	k	0.076	d^−1^
	B_0_	102.82	mL × g_vs_^−1^
	r	7.81	mL × g_vs_^−1^ × d^−1^
	R^2^	0.99	-
**IKW**	k	0.205	d^−1^
	B_0_	658.64	mL × g_vs_^−1^
	r	135.02	mL × g_vs_^−1^ × d^−1^
	R^2^	0.98	-
**IKW_SBC**	k	0.214	d^−1^
	B_0_	674.60	mL × g_vs_^−1^
	r	143.36	mL × g_vs_^−1^ × d^−1^
	R^2^	0.98	-

## Data Availability

All data derived during the experiments are given in the paper.

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
