# Peer review of "Enhanced Production of Biogas Using Biochar–Sulfur Composite in the Methane Fermentation Process"

_materials, 2022, doi:10.3390/ma15134517_

Round 1
Reviewer 1 Report
The authors of the studies have revealed a potential use of Sulphur Biochar Composites (SBC) to increase the yield of biogas in anaerobic digestion. Although the authors have mentioned the required qualitative study for establishing this concept. Overall this article provides some useful information on biogas production by using SBC, however, there are a few issues that need to be addressed. This article should be published after addressing the following general questions.
1. Agricultural and environmental groups are interested in determining the quantity and type of sulfur in biochars because of the potential use of biochar for soil augmentation, and plants need sulfur, which makes determining the form of sulfur in biochars of interest. Can you explain how your novel research will benefit circular economies? Do you plan to study the environmental impact or have you already done so? If yes, what factors will be most significant? Discuss them in your findings.
2. What are the effects of biochar production from different feedstocks and sulfur addition on biogas production? How do you plan to investigate potential feedstocks for more efficient SBC so that it would be more advantageous than existing methods in the future? Comparing your results in terms of application use of biochar or discussing it in the introduction section, based on sulfur content.
3. The proof-of-the-concept? What other findings motivated you to write these words in your title besides biogas production? Though qualitative research is still in the pipeline, is there any specific finding or property of SBC that is contributing to an increase of 4% in biogas production?
4. Lines 107-108; Explain this statement with references.
5. The number of experiments and results is limited. The significance of the conclusion is also limited, and the result cannot be explained from the perspective of the mechanism. Were these experiments performed in replicates? Or planning to have it in your next qualitative findings.
6. The title change is required. Biochar Sulphur Composite or SBC?
Author Response
The responses on the Reviewer comments are in the attached file.

Reviewer 2 Report
I suggest authors to consider points as mentioned in the attached comments file and revise accordingly.

Author Response

(The authors gave the same response as above.)

Reviewer 3 Report
This study proposed new idea regarding addition of sulfur-biochar composite to improve methane fermentation of organic waste. The composite consisted of 40% biochar and 60% of sulfur was added to the process. It was found that, taddition of 1% of the composite increased the biogas potential by 4%. Basically, some interesting results were reported. The whole manuscript is also well written. Overall, I think it can meet the standards of this journal, and I suggest a major revision for this work. The detailed comments are listed as follows:
1. Abstract: most of abstract is about the background, while only three sentences descript the experiments. It is unacceptable for a research study.
2. Line 34: This sentence should be supported by literature like Desalination 2013, 314, 169-188; Water Science and Technology 2014, 69(8), 1712-1719
3. Figure 2: The scale bar should be given.
4. Figure 3 should be drawn in standard form
5. Conclusions: no reference is allowed in this section.
6. Conclusions: remove the background introduction, and provide the experiment process and data.
Author Response

(The authors gave the same response as above.)

Round 2
Reviewer 1 Report
The authors have addressed all the comments.
Author Response
The response to the reviewer's comments is in the attached file.

Reviewer 2 Report
Overall manuscript has been revised and improved at certain extent, but still 2 points need to be checked and revised again.
1. In Reference no. 1, authors name Approach N should be deleted. Please check the names of authors etc. properly and include in the revised MS.
2. About Title, I suggest authors to use title something like this e.g. Enhanced production of biogas using biochar-sulfur composite in methane fermentation process.
Author Response

(The authors gave the same response as above.)
